# Differential impact of admission type and clinical complexity on diabetes hospitalization costs among African American and hispanic patients in Southeastern Virginia

Ismail El Moudden[ID]*, Asra Amidi, Reem Sharaf-Alddin, Michael C. Bittner, Qi Zhang

Old Dominion University, Norfolk, Virginia, United States of America

* elmoudi@odu.edu

## Abstract

### Background

Diabetes mellitus (DM) imposes substantial healthcare costs with documented disparities among African Americans and Hispanic patients. To inform care delivery and resource allocation, this study identified hospitalization cost predictors among African American and Hispanic patients with diabetes in Southeastern Virginia.

### Methods

We analyzed 6,011 hospital discharges from the Virginia Health Information database (2016–2020) for adults aged 18–85 with diabetes. Discharges were classified by Medicare Severity Diagnosis-Related Groups: DM with complications/comorbidities (DCC, n = 3,328), DM with major complications/comorbidities (DMCC, n = 1,518), and DM without major complications/comorbidities (DWO, n = 1,165). Because cost distributions were right-skewed (skewness 3.5–8.24), we used log-linear regression with robust standard errors and back-transformed coefficients to percentage changes.

### Results

Mean age differed by classification: DWO 38.7±17.2 years, DCC 47.4±17.4, DMCC 54.9±17.4. The cohort was predominantly African American (98.2–99.1%). For DWO, urgent admission was the strongest predictor, associated with 239.5% higher costs versus emergency admissions (95% CI, 220.8–258.2; p<0.001). Other significant predictors included skilled nursing facility discharge (SNF) (69.7–119.2% increase), primary procedures (11.0–53.8% increase), and peptic ulcer disease (66.1–135.8% increase. Readmission effects varied by classification: in univariable models, readmission was associated with 5.8% lower costs in DMCC (p<0.001); in multivariable models, this association attenuated and was not statistically significant (−3.5%; 95% CI, −9.0 to 2.3; p=0.230). By contrast, DCC and DWO showed increases of 13.7% and 6.0%, respectively.

**Data availability statement:** The data underlying this study cannot be shared publicly due to licensing agreements and patient privacy restrictions under the Health Insurance Portability and Accountability Act (HIPAA). The data are owned and licensed by Virginia Health Information (VHI), a third-party organization, and public sharing is prohibited under the terms of the license. Researchers who meet the criteria for access to confidential data may request access directly from Virginia Health Information (VHI) via https://www.vhi.org/pld.

**Funding:** This research was generously funded by the Hampton Roads Biomedical Research Consortium (HRBRC)- Project Number 958830-005, 2023.

**Competing interests:** The authors have declared that no competing interests exist.

**Abbreviations:** CI, Confidence Interval; DCC, Diabetes with Complications and Comorbidities; DKA, Diabetic Ketoacidosis; DM, Diabetes Mellitus; DMCC, Diabetes with Major Complications and Comorbidities; DWO, Diabetes without Major Complications or Comorbidities; EVMS, Eastern Virginia Medical School; HR, Hampton Roads; HRBRC, Hampton Roads Biomedical Research Consortium; LOS, Length of Stay; ML, Machine Learning; MS-DRG, Medicare Severity Diagnosis-Related Group; PCD, pulmonary circulation disorders; $R^2$, Coefficient of Determination; RATs, Readmissions and Transfers Supplemental Data S; SD, Standard Deviation; SNF, Skilled Nursing Facility; VHI, Virginia Health Information; VIF, Variance Inflation Factor.

## Conclusions

Admission type particularly urgent admissions among patients without major complications, was a key cost driver. Findings support risk stratification in all emergency departments, with priority in systems serving large proportions of minority patients. Heterogeneous readmission effects across classifications indicated the need for nuanced quality metrics. These results provided baseline data for predictive modeling to improve diabetes care and reduce disparities in minority populations.

## Introduction

Diabetes Mellitus (DM), a major chronic disease, presents a growing global health burden due to its long, difficult-to-manage course, leading to severe complications such as ischemic heart disease, stroke, peripheral vascular disease, renal failure, and irreversible consequences if uncontrolled such as diabetic foot [1]. These medical challenges are mirrored in their substantial societal impacts, including elevated healthcare costs, lost productivity, premature mortality, and diminished quality of life [2] Additionally, minority populations in the US experience worse DM-related health outcomes, likely influenced by various social determinants of health, including socioeconomic status, education level, geography, living conditions, and access to health care [3]. The convergence of clinical complexity and social disadvantage creates a convergence of clinical complexity and social disadvantage of escalating costs and deteriorating outcomes that demands urgent, data-driven intervention.

The American Diabetes Association in its 2022 Economic Report stated that the total annual cost was $412.9 billion, comprising $306.6 billion in direct health care cost and $106.3 billion in indirect costs such as caregiver burden or lost productivity [4]. This signifies that people diagnosed with DM now account for one of every four health care dollars spent in the United States (5). Medical expenses associated with DM are 2.6 times greater than what was expected for persons who do not have DM, highlighting the significant economic burden caused by the condition [5], in addition, there has been a 3% increase DM since 2017, resulting in a total rise of 14% compared to the published statistics in 2012. The medical expenses directly associated with DM have increased by 7% since 2017 and by 35% since 2012, indicating a projected rising pattern in the economic consequences of managing and treating DM [4]. These national trends likely underestimate the burden in minority communities where complications occur earlier and more severely.

DM disproportionately affects racial and ethnic minority groups in the United States, with African Americans experiencing significantly higher prevalence rates and worse health outcomes compared to non-Hispanic Whites [6,7]. These disparities are rooted in a complex interplay of genetic predisposition, environmental exposures, and socioeconomic inequality [8]. Factors such as limited access to healthy food, safe neighborhoods for physical activity, and quality healthcare services contribute to the increased burden of disease in these communities. Furthermore, implicit bias in the healthcare system, along with insurance and income

gaps, often results in delayed diagnoses and suboptimal management. As a result, minority populations are less likely to receive critical diabetes-related services, including Hemoglobin A1c (HbA1c) testing, annual cholesterol screenings, and retinal exams. These preventative measures are crucial for early detection, controlling the disease's progression, and preventing complications [9] Although expanding screening efforts has the potential to reduce health care cost and improve long-term outcomes, only 30% of African Americans have received recommended DM screenings [10,11,12]. This persistent gap between healthcare need and access not only perpetuates a cycle of preventable complications and escalating costs, but also leads to disproportionately high morbidity, mortality, and financial burden in communities of color.

In Virginia, diabetes impacts more than 733,000 individuals, with both prevalence and hospitalization rates increasing consistently in recent years [13]. From 2016 to 2020, prevalence rose from 10.4% to 11.1%, with hospitalization rates reflecting this increase. The burden is unevenly distributed; Black Virginians consistently face disproportionately elevated hospitalization rates relative to other racial and ethnic groups. State data indicates that the Eastern region, including Southeastern Virginia, carries a significant burden. This area exhibits socio-demographic diversity, with approximately 30% of residents identifying as African American, a population disproportionately impacted by costly diabetes complications.

Approximately 60% of adults with diabetes in Virginia are obese, exacerbating the risk of severe complications and expensive hospitalizations. The state-level patterns suggest that in the diverse communities of Hampton Roads (HR), especially among African Americans, the interplay of socioeconomic disadvantage and increased complication risk likely contributes to a substantial portion of high-cost diabetes care.

Building on these patterns, we aimed to identify predictors of diabetes-related hospitalization costs in Southeastern Virginia, focusing on African American By quantifying region-specific cost drivers, this study provides critical evidence to inform targeted care delivery strategies. The findings establish a foundation for predictive modeling approaches aimed at preventing high-cost hospitalizations and reducing disparities in diabetes outcomes. We aimed to identify sociodemographic, clinical, and administrative predictors of hospitalization costs among African American and Hispanic adults with diabetes in Southeastern Virginia.

## Methods

### Study design and setting

This retrospective cohort study analyzed hospital discharge data from African American and Hispanic patients with diabetes mellitus in Southeastern Virginia between January 1, 2016, and December 31, 2020. The study was designed to identify sociodemographic and clinical determinants of hospitalization costs among racial minorities, serving as the foundational analysis for a HR Biomedical Research Consortium (HRBRC) funded machine learning (ML) prediction platform.

### Data source

Data were obtained from the Virginia Health Information (VHI) database, a comprehensive repository containing records from all hospital discharges at licensed hospitals throughout Virginia. VHI represents the most complete source of inpatient utilization data in the Commonwealth, with mandatory reporting from all acute care facilities. The database captures detailed patient demographics, administrative records, clinical diagnoses and procedures, and financial information for each hospital discharge, providing a robust foundation for population-level analysis.

### Study population

We included all hospital discharges for African American and Hispanic patients aged 18–85 years with a primary or secondary diagnosis of diabetes mellitus, regardless of insurance status. Patients were identified using International

Classification of Diseases, 10th Revision, Clinical Modification (ICD-10-CM) codes for diabetes (E08–E13). Hispanic ethnicity was identified through hospital-reported ethnicity fields, while race was determined from standardized admission records. Discharges were classified into three mutually exclusive categories based on Medicare Severity Diagnosis-Related Group (MS-DRG) codes: DCC corresponding to MS-DRG 638–639, DMCC corresponding to MS-DRG 637, and DWO corresponding to MS-DRG 640. A complication or comorbidity was defined as a secondary diagnosis that directly impacts diabetes treatment and resource utilization, while major complications included conditions substantially increasing care complexity, such as diabetic ketoacidosis (DKA), hyperosmolar states, or severe organ involvement.

## Data collection and variables

The primary outcome was total hospital charges per discharge. While charges may differ from payments, they provide a consistent relative measure across patients and facilities; all results are interpreted as percentage differences in charges.

Demographic variables included age calculated from date of birth to admission date and expressed in years for adults, sex recorded as male or female, race with specific focus on African American identification, and patient ZIP code for geographic analysis. Clinical variables comprised length of stay (LOS) calculated as discharge date minus admission date, comorbidity burden quantified as the count of Elixhauser comorbidities identified using Agency for Healthcare Research and Quality software, and thirty-one specific comorbidities including congestive heart failure, valvular disease, pulmonary circulation (PCD) disorders, peripheral vascular disease, hypertension, paralysis, neurological disorders, chronic pulmonary disease, diabetes complications, hypothyroidism, renal failure, liver disease, peptic ulcer disease, AIDS/HIV, lymphoma, metastatic cancer, solid tumors, arthropathies, coagulopathy, obesity, weight loss, fluid and electrolyte disorders, blood loss anemia, deficiency anemia, alcohol abuse, drug abuse, psychoses, and depression. We also captured primary procedure presence or absence based on ICD-10 procedure codes and a complications indicator based on presence of ICD-10 codes T80–T88 indicating complications of medical care.

Administrative variables encompassed admission type categorized as emergency, urgent, elective, newborn, or trauma; admission source indicating point of origin including home, transfer from another hospital, or SNF; discharge destination coded as home, SNF, home health services, transfer to another facility, expired, or other; insurance type classified as Medicare, Medicaid, private or commercial, self-pay, or other; and readmission status identified through encrypted patient identifiers allowing longitudinal tracking. Those who were identified as being dual Medicare/Medicaid were kept separate from the sole Medicare and Medicaid patients, moved to the "Other" category within the insurance parameter. This was to ensure conclusions drawn from insurance were not impacted by some patients receiving multiple benefits from various insurance avenues. Geographic variables included Virginia health planning region and district codes, as well as county codes derived from patient ZIP codes.

## Data cleaning and quality assurance

Data cleaning followed VHI standards with additional study-specific procedures. Records were excluded if they contained fatal errors in admission date, discharge date, patient status at discharge, date of birth, principal diagnosis, or principal procedure. Duplicate records were identified and removed using VHI's unique record identifier system. For non-fatal errors, invalid entries were recoded to "unknown" categories to maximize data retention while maintaining integrity. Age values exceeding 120 years or negative values triggered manual review, while LOS values exceeding 365 days were verified against admission and discharge dates. Cost outliers beyond the 99.9th percentile were winsorized to reduce the influence of data entry errors while preserving legitimate high-cost cases. Missing data patterns were assessed for all variables, and complete case analysis was used for multivariable models as missing data comprised less than 2% for key variables. Sensitivity analyses using multiple imputation yielded similar results.

## Statistical analysis

Based on preliminary data showing mean log-transformed costs of 10.2 with a standard deviation of 1.8, we calculated that our sample sizes of 3,328 for DCC, 1,518 for DMCC, and 1,165 for DWO provided greater than 80% power to detect a 10% difference in costs between groups at $\alpha = 0.05$ using two-tailed tests, and greater than 90% power to detect clinically meaningful effect sizes with Cohen's $d$ exceeding 0.3 for primary predictors. Patient characteristics were summarized using means with standard deviations for continuous variables and frequencies with percentages for categorical variables. Distributions were assessed for normality using Shapiro–Wilk tests and visual inspection of histograms and Q–Q plots. Cost distributions demonstrated severe right skewness with values of 3.5 for DCC, 4.74 for DMCC, and 8.24 for DWO, necessitating logarithmic transformation for all regression analyses. Associations between individual predictors and log-transformed costs were examined using $t$-tests for binary categorical variables, one-way analysis of variance (ANOVA) for multi-category variables with Tukey's post-hoc comparisons, Pearson correlation coefficients for continuous variables, and simple linear regression to quantify effect sizes. All univariable results were back-transformed to percentage changes using the formula: Percentage Change = $(e^\beta - 1) \times 100\%$. Multivariable log-linear regression models were developed separately for each diabetes classification to identify independent predictors while controlling for confounding. The general model specification was $\log(\text{Charges}) = \beta_0 + \beta_1(\text{LOS}) + \beta_2(\text{Age}) + \beta_3(\text{Comorbidity Count}) + \beta_4(\text{Sex}) + \beta_5(\text{Primary Procedure}) + \beta_6(\text{Admission Type}) + \beta_7(\text{Insurance}) + \beta_8(\text{Discharge Destination}) + \beta_9(\text{Readmission}) + \beta_{10}(\text{Complications}) + \varepsilon$, where categorical variables were converted to indicator variables with reference categories of male for sex, absent for primary procedure, Medicare for insurance, home for discharge destination, emergency for admission type, and no for readmission.

Multicollinearity was assessed using variance inflation factors (VIF), with variables having VIF exceeding 10 removed iteratively, starting with the highest VIF, until all remaining predictors showed VIF less than 10. Final models showed all VIF values less than 5, indicating minimal multicollinearity. Given the hierarchical nature of the data with multiple admissions nested within patients, we employed Huber-White sandwich estimators to calculate robust standard errors, providing conservative confidence intervals(CI) that account for potential correlation within patients having multiple admissions and heteroscedasticity in the residuals.

Model diagnostics included residual plots to assess homoscedasticity and linearity assumptions, Cook's distance to identify influential observations using a threshold exceeding $4/n$, Durbin–Watson statistics to test for autocorrelation, condition indices to assess multicollinearity beyond VIF, and link tests to verify model specification. To enable comparison across predictors with different scales and distributions, we calculated standardized effect sizes from univariable analyses, ranked predictors by magnitude of percentage cost increase, categorized impacts as Extreme for increases exceeding 100%, Very High for 50–100%, High for 30–50%, or Moderate for 10–30%, and created visualizations combining point estimates with 95% CI.

Sensitivity analyses excluded 2020 data to assess COVID-19 pandemic effects, used median regression at the 50th percentile to verify findings were not driven by outliers, employed propensity score matching for key comparisons such as urgent versus emergency admissions, and tested interaction terms between age and comorbidity count as well as between diabetes classification and primary predictors. All analyses were performed using SAS version 9.4 (SAS Institute, Cary, NC) for data management and primary analyses, and R version 4.3.1 (R Foundation for Statistical Computing, Vienna, Austria) for advanced modeling and visualization. Specific R packages included tidyverse for data manipulation, ggplot2 for visualizations, sandwich for robust standard errors, and car for regression diagnostics. Statistical significance was set at $p < 0.05$ for all tests.

## Ethical considerations

The study was approved by the Institutional Review Board at Eastern Virginia Medical School (EVMS) with Institutional Review Board (IRB) approval number 23–05-NH-0118 and included a waiver of informed consent given the use of de-identified administrative data. Data were securely transferred via encrypted channels and stored on

password-protected devices within EVMS's Health Insurance Portability and Accountability Act compliant firewall. Access was restricted to authorized research team members who completed human subjects protection training. All analyses were conducted at the aggregate level with cell suppression for groups containing fewer than 11 patients to prevent potential re-identification. Results are reported in accordance with STROBE (Strengthening the Reporting of Observational Studies in Epidemiology) guidelines for observational studies.

## Results

### Study population characteristics

This study included 6,011 hospital discharges related to diabetes mellitus among African American and Hispanic patients in Southeastern Virginia between 2016 and 2020. Patients with diabetes with DCC comprised the majority (n = 3,328, 55.4%), followed by DMCC; n = 1,518, 25.3%) and (DWO; n = 1,165, 19.4%) (Table 1).

The demographic and clinical profiles varied significantly across diabetes classifications. Mean age increased with disease complexity: DWO patients were youngest at 38.7 ± 17.2 years, DCC patients averaged 47.4 ± 17.4 years, and DMCC patients were oldest at 54.9 ± 17.4 years (p < 0.001). The population was predominantly African American (98.2%–99.1%) with minimal Hispanic representation. Gender distribution was relatively balanced across groups, with females comprising 48.0% of DCC, 51.1% of DMCC, and 53.4% of DWO patients.

### Clinical complexity and healthcare utilization

Clinical complexity metrics showed expected gradients across classifications. Mean LOS increased progressively: 2.5 ± 1.6 days (DWO), 3.3 ± 2.7 days (DCC), and 5.0 ± 4.3 days (DMCC) (p < 0.001). Comorbidity burden showed similar patterns, with mean comorbidity counts of 1.7 ± 1.4 (DWO), 2.9 ± 1.6 (DCC), and 3.8 ± 1.8 (DMCC). Notably, 17.7% of DWO patients had no comorbidities, compared with 3.1% of DCC and 1.8% of DMCC patients. A majority of DMCC patients (56.7%) had four or more comorbidities.

Primary procedures were performed in 73.1% (DWO), 67.9% (DCC), and 74.4% (DMCC) of discharges. Emergency admissions predominated across all groups (94.1%–95.8%), whereas urgent admissions were less common (2.9%– 4.9%). Discharge disposition reflected severity: home discharge occurred in 84.3% (DWO), 71.7% (DCC), and 50.9% (DMCC); SNF discharge was DWO 0.7%, DCC 2.8%, DMCC 12.5%.

### Healthcare cost distribution patterns

Cost distributions exhibited severe right skewness that increased as clinical complexity decreased (Fig 1). Skewness values were 3.5 for DCC, 4.74 for DMCC, and 8.24 for DWO. Standardized distributions (Fig 1) revealed distinct patterns: DCC showed moderate variability with a stable central tendency; DMCC demonstrated the widest spread with prominent high-charge outliers; DWO displayed a compressed distribution with rare but extreme outliers. These patterns suggest that traditional classification may not capture all the factors driving resource use.

Standardized hospital charge distributions for diabetes with complications/comorbidities (DCC), diabetes with major complications/comorbidities (DMCC), and diabetes without major complications/comorbidities (DWO). Distributions were severely right-skewed, with skewness of 3.5 (DCC), 4.74 (DMCC), and 8.24 (DWO), indicating rare but extreme outliers among DWO discharges. Abbreviations: DCC, diabetes with complications/comorbidities; DMCC, diabetes with major complications/comorbidities; DWO, diabetes without major complications/comorbidities. Data source: Virginia Health Information, 2016–2020.

### Cost drivers stratified into three distinct impact tiers

Extreme impact factors (>100% increase) included only three predictors: urgent admission for DWO (239.5%), peptic ulcer disease in DMCC (135.8%, 95% CI: 36.5–307.1), and SNF discharge for DMCC (119.2%, 95% CI: 106.4–132.0).

**Table 1. Patient characteristics by DM type.**

| | DCC | DMCC | DWO |
|---|---|---|---|
| **n (%)** | n = 3,328 (55.37%) | n = 1,518 (25.25%) | n = 1,165 (19.38%) |
| Demographics | | | |
| Age, Year* | 47.4 ± 17.4 | 54.9 ± 17.4 | 38.7 ± 17.2 |
| African American† | 3,277 (98.5) | 1,505 (99.1) | 1,144 (98.2) |
| Female † | 1,596 (48.0) | 776 (51.1) | 622 (53.4) |
| Clinical Complexity | | | |
| Length of stay, days* | 3.3 ± 2.7 | 5.0 ± 4.3 | 2.5 ± 1.6 |
| Comorbidities* | 2.9 ± 1.6 | 3.8 ± 1.8 | 1.7 ± 1.4 |
| Comorbidity Count | | | |
| 0† | 103 (3.1) | 27 (1.8) | 206 (17.7) |
| 1† | 581 (17.5) | 120 (7.9) | 393 (33.7) |
| 2† | 867 (26.1) | 188 (12.4) | 294 (25.2) |
| 3† | 711 (21.4) | 322 (21.2) | 152 (13.1) |
| 4† | 539 (16.2) | 345 (22.7) | 75 (6.4) |
| 5 +† | 527 (15.8) | 516 (34.0) | 45 (3.9) |
| Primary procedure† | 2,259 (67.9) | 1,129 (74.4) | 851 (73.1) |
| Complications present† | 63 (1.9) | 55 (3.6) | 0 (0) |
| Discharge | | | |
| Home† | 2,387 (71.7) | 772 (50.9) | 982 (84.3) |
| Skilled Nursing Facility† | 92 (2.8) | 189 (12.5) | 8 (0.7) |
| Home health† | 673 (20.2) | 411 (27.1) | 90 (7.7) |
| Other† | 176 (5.3) | 146 (9.6) | 85 (7.3) |
| Insurance Type | | | |
| Medicare† | 1,042 (31.3) | 738 (48.6) | 192 (16.5) |
| Medicaid† | 671 (20.1) | 253 (16.7) | 301 (25.8) |
| Self-pay† | 574 (17.3) | 146 (9.6) | 275 (23.6) |
| Other† | 1,041 (31.3) | 381 (25.1) | 397 (34.1) |
| Admission Type | | | |
| Emergency† | 3,130 (94.1) | 1,454 (95.8) | 1,108 (95.1) |
| Urgent† | 163 (4.9) | 44 (2.9) | 50 (4.3) |
| Elective† | 33 (1.0) | 14 (0.9) | 6 (0.5) |
| Readmission | | | |
| Yes† | 1,055 (31.7) | 595 (39.2) | 329 (28.2) |

Note: DCC = Diabetes with Complications and Comorbidities; DMCC = Diabetes with Major Complications and Comorbidities; DWO = Diabetes without Major Complications or Comorbidities; †: n(%); *:Mean±SD

Very high impact factors (50–100% increase) encompassed primary procedures for DMCC (87.4%), SNF discharges for DCC (81.7%) and DWO (69.7%), peptic ulcer disease in DCC (66.1%), and presence of complications in DCC (52.3%). High impact factors (30–50% increase) included various comorbidities and discharge destinations, with 17 additional predictors showing increases between 15% and 50%.

### Ranked healthcare cost drivers

Comprehensive analysis of cost drivers revealed dramatic variations in impact magnitude across 25 key predictors (Table 2). The single most powerful cost driver was urgent admission for DWO patients, producing an unprecedented

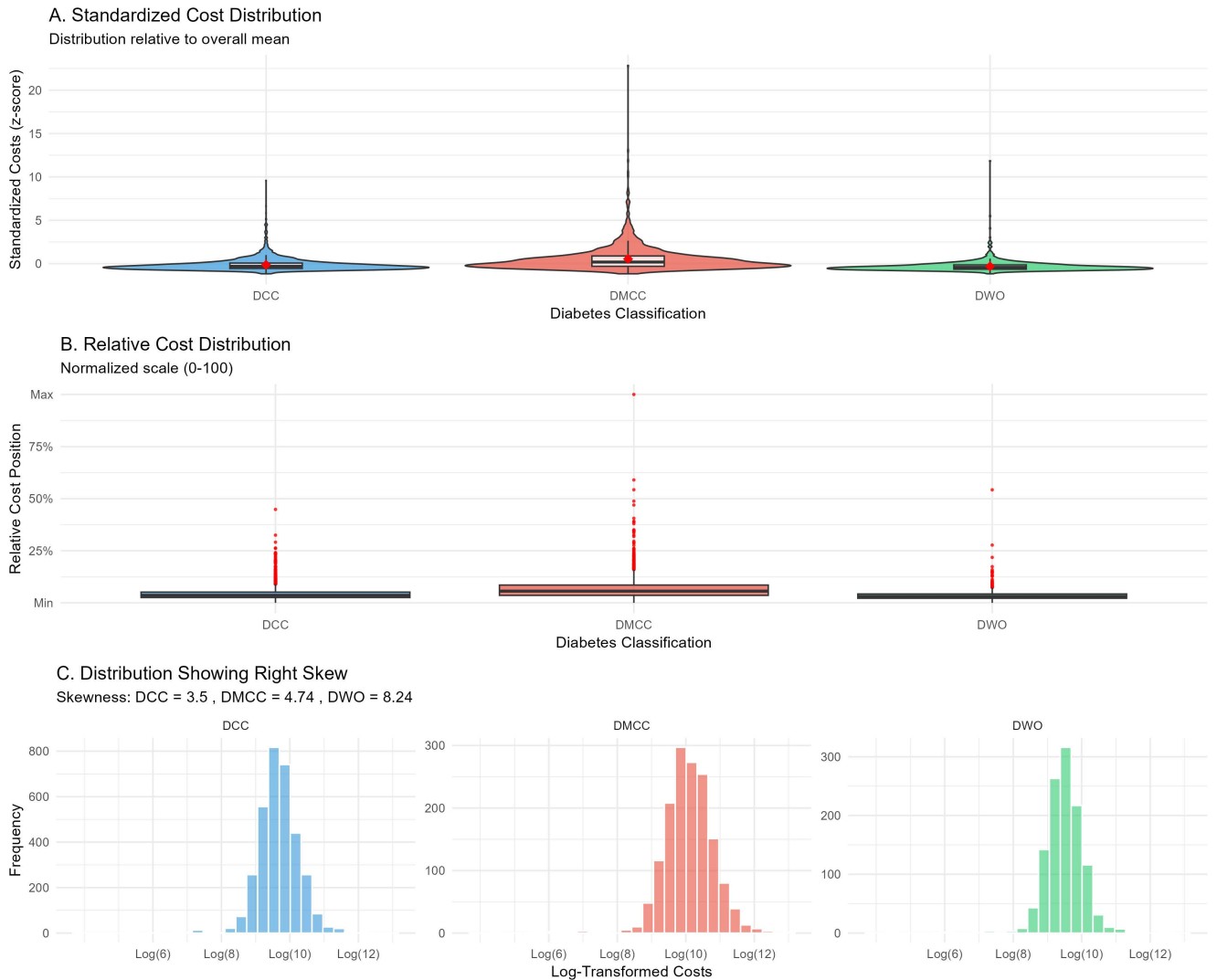

**Fig 1. Cost distributions by diabetes classification.**

239.5% increase in costs compared to emergency admissions (95% CI: 220.8–258.2; p < 0.001). This effect, which represents nearly 2.5 times baseline costs, exceeded all other predictors including major procedures and complex comorbidities.

### Differential effects across diabetes classifications

The forest plot (Fig 2) demonstrated heterogeneous effects by classification. Urgent admission showed an extreme effect only in DWO, with more moderate effects in DCC (16.1%) and DMCC (20.0%). Peptic ulcer disease increased charges by 135.8% in DMCC and 66.1% in DCC. SNF discharge showed a graded pattern aligned with baseline complexity: DWO 69.7%, DCC 81.7%, DMCC 119.2%.

Percent change in hospital charges (95% CI) from univariable log-linear models for key predictors, estimated separately within DCC, DMCC, and DWO. Reference categories were: emergency (admission type), home (discharge destination), none (primary procedure and complication), Medicare (insurance), and male (sex). The plot shows heterogeneity

**Table 2. Detailed comorbidities' impact on hospital charges by diagnosis type.**

| Comorbidity | DCC | DMCC | DWO |
|---|---|---|---|
| Peptic Ulcer Disease | 66.1%** (19.9%, 130.2%) | 135.8%** (36.5%, 307.1%) | — |
| Blood Loss Anemia | 45.7%** (14.8%, 85%) | — | — |
| Pulmonary Circulation Disorders | 38%** (12.1%, 69.9%) | 44.9%*** (16.8%, 79.8%) | — |
| Coagulopathy | 42.7%*** (22.7%, 65.8%) | 36.7%*** (18.2%, 58%) | — |
| Valvular Disease | 49%*** (23.7%, 79.2%) | 15.4%* (−4.4%, 39.3%) | — |
| Liver Disease | 15.7%** (5.5%, 26.9%) | 28.2%*** (13.1%, 45.3%) | 21.4%* (1.9%, 44.6%) |
| Lymphoma | 20.9%** (5.2%, 38.9%) | — | — |
| Weight Loss | 9.1% (−0.3%, 19.4%) | 18.7%** (7.7%, 30.9%) | 25%* (3.5%, 51%) |
| Neurological Disorders | 28.6%*** (17.7%, 40.6%) | 16.5%** (6.4%, 27.5%) | 7.5% (−4.2%, 20.6%) |
| Chronic Pulmonary Disease | 28.6%*** (17.7%, 40.6%) | 16.5%** (6.4%, 27.5%) | 7.5% (−4.2%, 20.6%) |
| Deficiency Anemia | 23.5%**** (17.1%, 30.3%) | 3.7% (−3.9%, 12.3%) | 21.3%**** (11.1%, 32.3%) |
| Arthropathies | 11.3% (−2.1%, 26.6%) | 15.6%* (−3.8%, 45.5%) | 17.1% (10.1%, 52.5%) |
| Hypertension | 16.2%*** (10.6%, 22%) | 15.1%** (4.3%, 27.1%) | 0.9% (−5.2%, 7.5%) |
| Congestive Heart Failure | 18.7%*** (11.9%, 26%) | 13.1%*** (7.3%, 19.4%) | −1.5% (−3.6%, 0.7%) |
| Fluid/Electrolyte Disorders | 0.7% (−3.8%, 5.3%) | 21%*** (11.4%, 31.1%) | 3.6% (−2.7%, 10.4%) |
| Alcohol Abuse | 12.1%** (4.7%, 20.1%) | 13.3% (−1.9%, 30.7%) | −1.7% (−13.3%, 11.1%) |
| Psychoses | 3.5% (−3.7%, 11.3%) | −2.2% (−14.7%, 12.6%) | 21%** (5%, 39.5%) |
| Depression | 3.5% (−3.7%, 11.3%) | −2.2% (−14.7%, 12.6%) | 21%** (5%, 39.5%) |
| Peripheral Vascular Disease | 25.4%** (10.1%, 42.8%) | −11.8% (−28.2%, 8.4%) | 8.6% (−19.7%, 46.9%) |
| Drug Abuse | 4.9% (−2.2%, 12.5%) | 9.4% (−3.2%, 23.3%) | 7.7% (−1.2%, 17.4%) |
| Metastatic Cancer | 12.7% (−12.7%, 45.4%) | 1.5% (−25.8%, 38.8%) | — |
| Paralysis | 4.7% (−10.3%, 22.1%) | 7.7% (−11.7%, 31.5%) | — |
| Hypothyroidism | 7.1% (−5.2%, 20.9%) | 3.9% (−8.7%, 18.7%) | 5.9% (−13%, 28.8%) |
| Obesity | 4.7%* (0.3%, 9.3%) | 3% (−5.8%, 12.8%) | 3.2% (−8%, 15.7%) |
| Renal Failure | 9.8%*** (4.5%, 15.4%) | 5.3%* (−2.3%, 13.4%) | −5.1% (−15.9%, 7.1%) |
| Solid Tumor | 1.6% (−15.6%, 22.3%) | 2.9% (−17.7%, 28.6%) | — |
| AIDS/HIV | — | −21.1%* (−36.2%, −2.4%) | — |

Note: DCC = Diabetes with Complications/Comorbidities; DMCC = Diabetes with Major Complications/Comorbidities; DWO = Diabetes without Major Complications/Comorbidities; CI = Confidence Interval. Values are percent differences in charges relative to the stated reference category; 95% CIs are shown in brackets. Asterisks denote significance thresholds (*** p < 0.001; ** p < 0.01; * p < 0.05). Cells marked "—" indicate not estimated or not applicable.

across classifications (e.g., urgent admission is extreme in DWO and more moderate in DCC/DMCC. Abbreviations: CI, confidence interval; DCC, diabetes with complications/comorbidities; DMCC, diabetes with major complications/comorbidities; DWO, diabetes without major complications/comorbidities; SNF. Data source: Virginia Health Information, 2016–2020.

## The readmission paradox

Readmission patterns defied conventional expectations. While readmission increased costs for DCC patients by 13.7% (95% CI, 11.9–16.2; p < 0.001) and for DWO patients by 6.0% (95% CI, 3.9–8.1; p < 0.001), DMCC patients demonstrated a paradoxical 5.8% cost reduction with readmission (95% CI, −8.0 to −3.6; p < 0.001). This counterintuitive finding persisted even in multivariable analysis, though it lost statistical significance after adjustment (−3.5%; 95% CI, −9.0–2.3; p = 0.230),

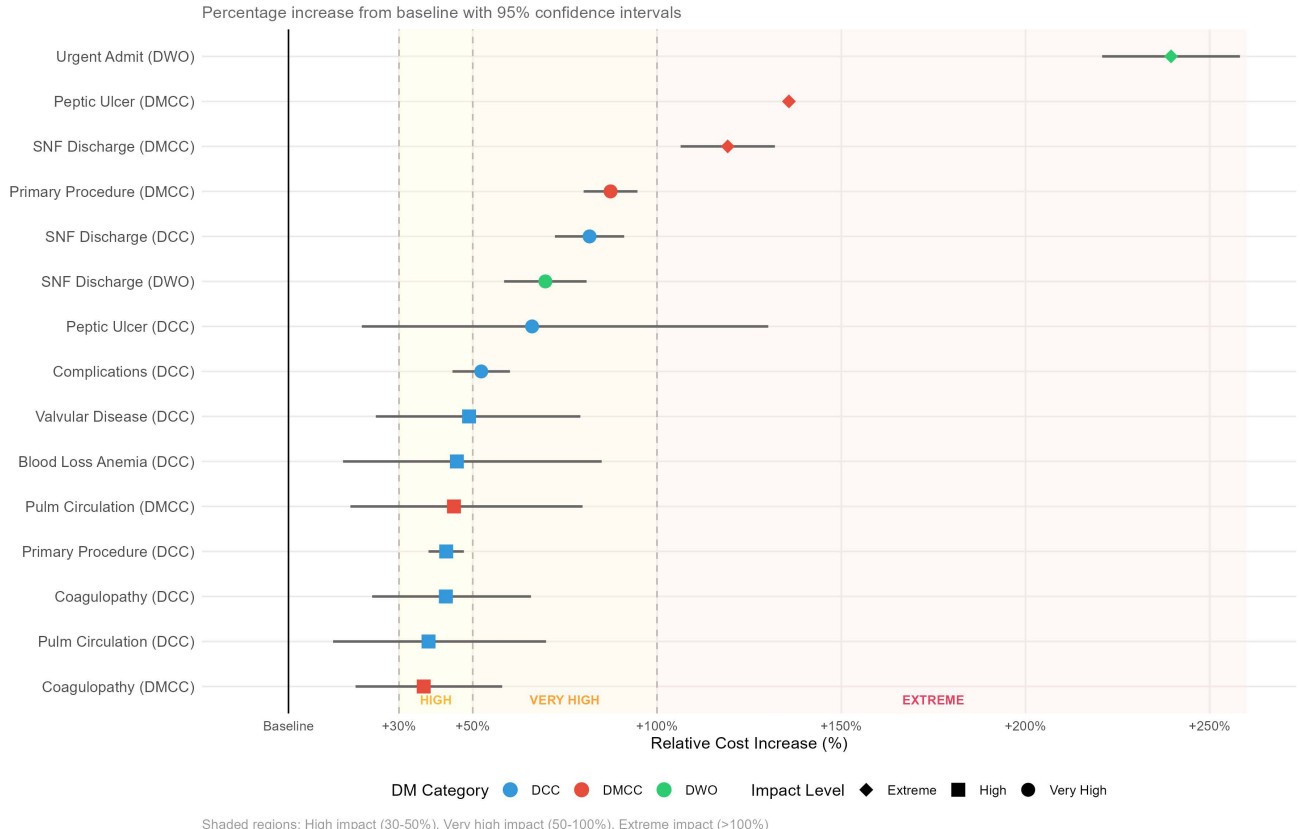

**Fig 2. Forest plot of cost predictors across diabetes classifications.**

suggesting partial mediation by measured covariates. One possible explanation is that DMCC patients incurred substantially higher costs during their initial admission, which may have reduced the intensity of services required at readmission.

### Insurance and Administrative Factors

Insurance patterns varied significantly by diabetes classification. Medicare covered 48.6% of DMCC, 31.3% of DCC, and only 16.5% of DWO patients, reflecting age and disability distributions. Self-pay was most common in DWO (23.6%) and least common in DMCC (9.6%). In univariable analysis, self-pay showed dramatic cost reductions compared to private insurance for DCC patients (−63.9%, $p < 0.001$) but minimal effect for DMCC (−3.5%, p = 0.056). These effects are substantiated in multivariable models.

### Admission type profoundly influenced costs

Beyond the extreme effect of urgent admissions for DWO, urgent admissions increased costs by 16.1% for DCC and 20.0% for DMCC compared to emergency admissions. Paradoxically, elective admissions showed lower costs than emergency admissions for DCC (−9.5%, p < 0.001) and DMCC (−18.0%, p < 0.001) but higher costs for DWO (9.3%, p < 0.001).

### Multivariable regression analysis

After adjusting for confounders, multivariable models identified consistent predictors while revealing important effect modifications (Table 3). LOS remained the strongest continuous predictor across all classifications, with each additional day

Table 3. **Multivariable log-linear models for hospitalization costs by diabetes classification (DCC, DMCC, DWO).**

| Predictor | DCC | DMCC | DWO |
|---|---|---|---|
| Length of Stay | 11.0%*** [8.0%, 14.1%] | 9.0%*** [6.6%, 11.5%] | 17.4%*** [14.6%, 20.2%] |
| Primary Procedure (vs. None) | 33.0%*** [27.5%, 38.8%] | 53.8%*** [43.4%, 64.9%] | 11.0%** [2.7%, 20.1%] |
| Comorbidity Count | 3.4%*** [2.3%, 4.5%] | 3.0%** [1.0%, 5.1%] | 2.7%** [0.9%, 4.5%] |
| Age | −0.2%** [−0.3%, −0.03%] | −0.5%*** [−0.7%, −0.2%] | −0.2%* [−0.5%, −0.03%] |
| Female (vs. Male) | −4.1%** [−7.5%, −0.7%] | −2.5% [−8.0%, 3.2%] | −4.1% [−9.6%, 1.7%] |
| SNF Discharge (vs. home) | 10.9% [−2.0%, 25.6%] | 16.6%*** [5.4%, 28.9%] | 5.8% [−1.8%, 36.1%] |
| Insurance (vs. Medicare) | | | |
| Medicaid | −1.4% [−7.6%, 5.1%] | 4.1% [−7.9%, 17.7%] | −3.5% [−15.1%, 9.7%] |
| Self-Pay | −5.0% [−11.2%, 1.7%] | 1.4% [−12.5%, 17.5%] | −12.7%* [−23.0%, −0.9%] |
| Other | −2.2% [−8.8%, 4.9%] | 3.6% [−9.1%, 18.1%] | −12.4%* [−23.0%, −0.3%] |
| Admission Type (vs. emergency) | | | |
| Urgent | 14.9%*** [12.1%, 17.7%] | 18.1%*** [14.7%, 21.5%] | 122.3%*** [113.6%, 131.0%] |
| Elective | −10.0%*** [−13.1%, −6.9%] | −19.9%*** [−24.3%, −15.5%] | 8.8%*** [5.9%, 11.7%] |
| Complication (vs.None) | 41.2%*** [37.3%, 45.1%] | 26.1%*** [22.6%, 29.6%] | — |
| Readmission | 2.9% [−0.93%, 6.9%] | −3.5% [−9.0%, 2.3%] | 3.0% [−2.65%, 9.0%] |
| Yes (vs. No) | 0.779 | 0.789 | 0.688 |
| Model R² | 11.0%*** [8.0%, 14.1%] | 9.0%*** [6.6%, 11.5%] | 17.4%*** [14.6%, 20.2%] |

Note: DCC = Diabetes with Complications/Comorbidities; DMCC = Diabetes with Major Complications/Comorbidities; DWO = Diabetes without Major Complications/Comorbidities; SNF = Skilled Nursing Facility; CI = Confidence Interval. Values are percent differences in cost relative to the stated reference category; 95% CIs are shown in brackets. Estimates are from multivariable log-linear regression with robust standard errors, fitted separately within each diabetes classification. Costs were log-transformed and back-transformed as % change = $(e^\beta − 1) \times 100$. Significance is denoted by asterisks (*** p < 0.001; ** p < 0.01; * p < 0.05); use roman 'p'.

increasing costs by 11.0% for DCC (95% CI, 8.0–14.1), 9.0% for DMCC (95% CI, 6.6–11.5), and 17.4% for DWO (95% CI, 14.6–20.2) (all p < 0.001). The larger effect in DWO patients suggested that extended stays in this group represent clinical deterioration requiring intensive intervention.

Primary procedures maintained strong independent effects: 33.0% increase for DCC (95% CI, 27.5–38.8), 53.8% for DMCC (95% CI, 43.4–64.9), and 11.0% for DWO (95% CI, 2.7–20.1) (all p < 0.01). Critically, urgent admissions retained their extreme impact for DWO patients even after full adjustment (122.3%, 95% CI, 113.6–131.0; p < 0.001), confirming this as the most robust predictor identified.

Age demonstrated small but significant negative associations with costs after adjustment: −0.2% per year for DCC (p = 0.02), −0.5% for DMCC (p < 0.001), and −0.2% for DWO (p = 0.026). This counterintuitive finding likely reflects survival bias and competing mortality risks in older patients. Comorbidity count showed expected positive associations: 3.4% per additional comorbidity for DCC, 3.0% for DMCC, and 2.7% for DWO (all p < 0.05).

Sex effects varied by classification. Females had 4.1% lower costs in DCC (p = 0.02) but showed no significant differences in DMCC or DWO after adjustment. SNF discharge remained significant only for DMCC patients (16.6% increase, p < 0.001) after adjustment, while showing trends for other groups.

## Sensitivity analyses

Exclusion of 2020 data to assess COVID-19 pandemic effects showed minimal impact on primary findings. 729 discharge records from 2020 were excluded in total, including 410 DCC patients, 207 DMCC patients, and 114 DWO patients. The

urgent admission effect for DWO patients remained extreme (235.2% vs. 239.5%), readmission patterns persisted, and model $R^2$ values changed by less than 0.02. This stability suggests our findings reflect underlying structural factors rather than pandemic-related disruptions.

## Model performance and predictive capacity

The final multivariable models achieved robust explanatory power with adjusted $R^2$ values of 0.779 for DCC, 0.789 for DMCC, and 0.688 for DWO. F-statistics ranged from 298.45 to 412.53 (all $p < 0.001$), confirming strong model fit. The lower $R^2$ for DWO patients (0.688) reflected the extreme variability in this group, where rare critical intervention points (urgent admissions) dramatically influenced costs. These baseline $R^2$ values, already explaining 69–79% of cost variance using administrative and clinical data alone, provide strong foundation for ML enhancement anticipated to achieve $R^2 > 0.85$ through incorporation of social determinants and unstructured clinical notes.

Sensitivity Analyses. Exclusion of 2020 data to assess COVID-19 pandemic effects showed minimal impact on primary findings. The urgent admission effect for DWO patients remained extreme (235.2% vs. 239.5%), readmission patterns persisted, and model $R^2$ values changed by less than 0.02. This stability suggests our findings reflect underlying structural factors rather than pandemic-related disruptions.

## Discussion

This comprehensive analysis of 6,011 diabetes-related hospitalizations among African American and Hispanic populations in Southeastern Virginia reveals paradigm-shifting insights into healthcare cost drivers. Some of these factors challenge conventional understanding of diabetes complexity and resource utilization making them especially noteworthy. Three findings fundamentally alter our approach to diabetes cost prediction and management. First, urgent admissions for patients with diabetes without major complications or comorbidities (DWO) triggered the highest cost increases among all factors analyzed, with an unprecedented 239.5% increase compared to emergency admissions, exceeding even major procedures and complex comorbidities. Second, readmission paradoxically reduced costs by 5.8% for the most complex patients with DMCC, suggesting that planned readmission strategies may be associated with lesser costs than prolonged initial stays. Third, our minority population develops major complications at substantially younger ages, with DMCC patients averaging 54.9 years and DWO patients only 38.7 years, indicating earlier onset of complications observed in this cohort requiring more aggressive intervention strategies.

The extraordinary cost impact of urgent admissions for patients with diabetes without major complications or comorbidities, representing nearly 2.5 times baseline costs, represents the most striking finding of our analysis and likely reflects acute metabolic emergencies such as DKA or severe hypoglycemia requiring intensive intervention. That these critical intervention points occur in patients without documented major complications reveals failures in our healthcare delivery system. Young adults presenting urgently with diabetes lacking major complications (DWO) may have inconsistent insurance coverage, medication access challenges, or limited health literacy, which are social determinants that our current classification systems fail to capture. This finding resonates with previous research demonstrating that diabetes leads to numerous health consequences requiring longer healthcare stays and more extensive procedures [14,15,16]. Our data suggest these consequences manifest most dramatically in patients who appear least complex by traditional measures. The concentration of these extreme costs in urgent rather than emergency presentations suggest a window of deterioration where patients recognize distress less have not yet reached crisis, representing a critical intervention opportunity currently being missed.

The counterintuitive finding that readmission reduces costs for DMCC patients challenges fundamental assumptions about healthcare quality metrics. While readmission increased costs for patients with diabetes with complications and comorbidities (DCC) by 13.7% and DWO patients by 6.0%, consistent with traditional expectations, the 5.8% reduction for DMCC patients suggests sophisticated care coordination rather than system failure. This aligns with research by Rubin

showing that hospital readmission patterns in diabetes are more nuanced than simple quality failures [17] and Dungan's findings that diabetes readmissions, especially for the African American and Hispanic population, require careful interpretation [18]. For patients with multiple major complications, attempting comprehensive treatment in a single admission may be neither feasible nor cost-effective. Staged procedures, interval optimization between interventions, or planned reassessments may represent evidence-based strategies that our current penalty-focused readmission metrics fail to recognize. This finding is particularly relevant for the African American and Hispanic population, who as noted by Jiang et al., are more likely to be readmitted, yet our data suggest this may sometimes represent appropriate care rather than disparity [19].

The age distribution across diabetes categories reveals alarming health equity implications that demand immediate attention. Our findings show that DWO patients average just 38.7 years, nearly two decades younger than typical diabetes major complication onset. Meanwhile DMCC patients at 54.9 years are approximately 7–10 years younger than national averages, confirming earlier onset of complications in minority populations. Literature indicates that diabetes prevalence interacts with sex at a hormonal level [20] while hospital utilization patterns also vary by demographic factors [21], and our data support this with differential patterns across classifications. These findings add nuance to the literature on sex-based differences in diabetes outcomes by demonstrating that while demographic factors may influence diabetes healthcare costs in simple analyses, these effects are largely attenuated when clinical factors are considered.

The interaction between age and diabetes severity was proven significant by Alrashed et al. who examined sex-specific associations and how they impact diabetes risk [22], with their findings indicating that males and females react differently on a physiological level and therefore need to be considered differently when identifying diabetes risk. Our results extend this understanding by showing that in minority populations, these sex-based differences intersect with age to create unique cost patterns. While looking at age by individually does not show large shifts between costs in diabetes types, when considering other parameters in conjunction with age, deviations in cost arise. Overall, the concentration of middle-aged adults across all diagnosis categories, with the highest severity in the youngest groups, is consistent with the global trend of increasing diabetes prevalence in younger populations [23,24]. This suggests this trend is accelerated in minorities, emphasizing the need for early intervention and management strategies targeted specifically at young minority adults.

When examining the broader context of our findings, it becomes clear that diabetes is impacting a younger minority population, with those under 65 showing the highest rates of diagnosis in all categories. This could indicate that the nutritional transition towards a high-calorie diet and sedentary lifestyle in the last few decades has disproportionately affected minority communities at younger ages [25]. Healthcare costs were also shown to be influenced by age in instances of diabetes with comorbidities, consistent with previous studies showing that the addition of underlying factors that develop with age exacerbates diabetes risk and therefore costs [5,26,27]. Utilizing national records, it becomes clear that age is a key component of both diabetes risk and complexity of care, resulting in higher healthcare costs [5]. This connection was even more apparent when breaking down age-adjusted prevalence by race, showing non-Hispanic black adults and adults of Hispanic ethnicity having higher rates than their White counterparts [26].

Type of insurance emerged as one of the main predictors of healthcare costs to the patient, though its effects were complex and varied by diabetes classification. Change of health insurance type or coverage is very common due to several factors such as shifting from private insurance provided by a workplace to Medicare for those older than 65 [28]. The predominant use of Medicare by those with DCC/DMCC, while DWO patients relied more on Medicaid and private insurance, matches other studies examining healthcare utilization patterns in diabetes [28,29]. This finding suggests a potential correlation between insurance type and diabetes severity, supported by research from Casagrande, Park, Herman, and Bullard [29] who noted a higher prevalence of Medicare and Medicaid/other public insurance among adults with diabetes compared to those without the condition.

Interestingly, while insurance appeared significant in univariable analysis, its impact diminished in multivariable models, suggesting confounding by age and severity. The exception was DWO patients, where self-pay was associated with

12.7% lower costs even after adjustment, possibly reflecting decisions to forego recommended treatments, leave against medical advice when facing financial constraints, or the receipt of hospital billing discounts. Age impacts numerous aspects of a person's healthcare profile, including the insurance they use, and it has been noted that upon reaching age 65 there are inherently higher healthcare costs for Medicare coverage, resulting in worse adherence to diabetes care [30].

The presence of comorbidities, particularly as the number increases, further exacerbates the healthcare costs of diabetes, aligning with studies quantifying the financial impact of specific comorbidities like cardiovascular disease [30]. Consistent with previous research, our data underscore the significant association between disease complexity and escalating healthcare costs. We found that patients with diabetes experience complications such as hypertension and fluid and electrolyte disorders in more than half of cases. As the number of comorbidities increased, the healthcare costs incurred increased proportionally, aligning with findings from Top et al. [31] and Stokes et al [32] who both reported increased costs associated with higher numbers of diabetes-related complications and multimorbidity combinations.

When examining specific comorbidities' impact on costs, several stand out in Southeastern Virginia, including peptic ulcer disease showing a 135.8% increase for DMCC patients, blood loss anemia, PCD, and coagulopathy. This resonates with other research noting diabetes connections to excessive blood loss [33], peptic ulcers [34] and pulmonary disorders [35]. The differential impact of these comorbidities across diabetes classifications suggests that traditional approaches to comorbidity adjustment may inadequately capture their true cost implications.

Among patients in Southeastern Virginia with diabetes, many required procedures during their hospital visits, with diabetes patients who had a procedure incurring significantly higher healthcare costs regardless of diagnosis category. The impact was most pronounced for DMCC patients (53.8% increase in adjusted models), reflecting the complexity of intervening in patients with multiple major complications. Diabetes leads to numerous health consequences requiring longer healthcare and more extensive procedures (14, 15, 16), and this was reflected in higher healthcare costs for patients with DCC and DMCC in Southeastern Virginia. This finding parallels observations by Hex et al. in the United Kingdom where only 40% of costs related to direct diagnosis and treatment while the remaining 60% were indirect costs derived from other health issues arising due to their diabetes diagnosis [36].

Administrative information revealed critical insights about how the circumstances surrounding a hospital visit independently influence costs. Admissions in general are higher for those suffering from diabetes due to the variety of additional ailments that coincide with it [37]. Our research indicated significant cost shifts for urgent and elective admissions compared to emergency visits, with urgent admissions showing the most dramatic effects. Friel et al. found admission type to be significant as well when comparing unscheduled emergency admissions to elective admissions, showing increased costs for elective readmissions [38].

Discharge destinations emerged as a powerful predictor, with those sent home consistently paying less than those transferred to different facilities or sent to other care facilities. Southeastern Virginia shows consistent significance in this pattern, aligning with Friel et al.'s observations that those sent to nursing homes or hospice care paid more overall (38). The extreme impact of SNF discharge for DMCC patients (119.2% increase) suggests that post-acute care represents a major cost driver requiring targeted intervention strategies.

These findings provide the essential foundation for our HR Biomedical Research Consortium-funded hospitalization cost prediction platform. The robust baseline predictive performance with $R^2$ values of 0.779 for DCC, 0.789 for DMCC, and 0.688 for DWO confirms the feasibility of achieving $R^2$ exceeding 0.85 through ML enhancement. The ranked cost drivers analysis (Table 4) establishes clear feature importance for algorithm development, with urgent admission status, discharge destination, and specific comorbidity combinations requiring priority in model architecture. The extreme variance in DWO costs particularly highlights opportunities for predictive intervention: identifying young adults at risk for metabolic crisis before urgent presentation could prevent both human suffering and financial catastrophe.

A monitoring dashboard could track these high-impact factors in real-time, enabling proactive intervention when risk patterns emerge. By incorporating the cost drivers identified here, from urgent admission patterns to specific comorbidity

**Table 4. Healthcare cost drivers ranked by magnitude of impact sociodemographic and clinical predictors stratified by effect size across diabetes classifications.**

| Rank | Impact Level | Cost Driver | DM Category | % Increase (95% CI) | Reference Category |
|---|---|---|---|---|---|
| 1 | EXTREME (>100%) | Urgent Admission | DWO | **239.5*** (220.8-258.2)** | Emergency |
| 2 | | Peptic Ulcer Disease | DMCC | **135.8*** (36.5-307.1)** | No peptic ulcer |
| 3 | | SNF Discharge | DMCC | **119.2*** (106.4-132.0)** | Home |
| 4 | VERY HIGH (50-100%) | Primary Procedure | DMCC | 87.4*** (80.1-94.7) | No procedure |
| 5 | | SNF Discharge | DCC | 81.7*** (72.3-91.1) | Home |
| 6 | | SNF Discharge | DWO | 69.7*** (58.5-80.9) | Home |
| 7 | | Peptic Ulcer Disease | DCC | 66.1** (19.9-130.2) | No peptic ulcer |
| 8 | | Complications Present | DCC | 52.3*** (44.5-60.1) | No complications |
| 9 | HIGH IMPACT (30–50% increase) | Valvular Disease | DCC | 49.0*** (23.7-79.2) | No valvular disease |
| 10 | | Blood Loss Anemia | DCC | 45.7** (14.8-85.0) | No anemia |
| 11 | | Pulmonary Circulation | DMCC | 44.9*** (16.8-79.8) | No PCD |
| 12 | | Coagulopathy | DCC | 42.7*** (22.7-65.8) | No coagulopathy |
| 13 | | Primary Procedure | DCC | 42.8*** (38.0-47.6) | No procedure |
| 14 | | Pulmonary Circulation | DCC | 38.0** (12.1-69.9) | No PCD |
| 15 | | Coagulopathy | DMCC | 36.7*** (18.2-58.0) | No coagulopathy |
| 16 | | Home Health | DCC | 34.9*** (31.7-38.1) | Home |
| 17 | | Complications Present | DMCC | 33.3*** (27.8-38.8) | No complications |
| 18 | MODERATE IMPACT (10–30% increase) | Other Discharge | DMCC | 28.2*** (24.7-31.7) | Home |
| 19 | | Liver Disease | DMCC | 28.2*** (13.1-45.3) | No liver disease |
| 20 | | Neurological Disorders | DCC | 28.6*** (17.7-40.6) | No neuro disorders |
| 21 | | Chronic Pulmonary | DCC | 28.6*** (17.7-40.6) | No COPD |
| 22 | | Peripheral Vascular | DCC | 25.4 *** (10.1-42.8) | No PVD |
| 23 | | Weight Loss | DWO | 25.0** (3.5-51.0) | No weight loss |
| 24 | | Deficiency Anemia | DCC | 23.5*** (17.1-30.3) | No anemia |
| 25 | | Home Health | DMCC | 22.1*** (18.8-25.4) | Home |

Note: DCC = Diabetes with Complications/Comorbidities; DMCC = Diabetes with Major Complications/Comorbidities; DWO = Diabetes without Major Complications/Comorbidities; **SNF** = Skilled Nursing Facility; PCD = Pulmonary Circulation Disorders; PVD = Peripheral Vascular Disease; COPD = Chronic Obstructive Pulmonary Disease. Values are percent differences in cost relative to the stated reference category; **95% CIs are shown in brackets**. Estimates are from **univariable log-linear regression with robust standard errors**, fitted separately within each DM classification. Costs were log-transformed to address right-skewness (skewness: DCC = 2.89; DMCC = 3.51; DWO = 4.20) and back-transformed as **% change = (e^β − 1) × 100**. Rankings are based on point estimates within each impact tier. **Significance:** ***p < 0.001; *p < 0.01; p < 0.05.

combinations, predictive models could identify high-risk patients before crisis onset. The strong baseline model performance demonstrates that enhanced prediction through ML is not just feasible but imperative for addressing healthcare disparities in minority populations with diabetes

This study's strengths include the comprehensive VHI database capturing all regional hospitalizations, robust statistical methodology with appropriate transformations for severely skewed cost data and focus on an understudied but critically important population. The ranked cost driver analysis provides an innovative visualization enabling rapid identification of intervention priorities. By examining administrative and clinical factors simultaneously, we uncovered interactions invisible to traditional analyses, particularly the extreme impact of urgent admissions in patients without significant complications.

Several limitations warrant consideration. Administrative data cannot capture crucial unmeasured confounders including medication adherence, health literacy, social support, and neighborhood-level factors that likely mediate the observed disparities. Our HRBRC platform will address this through electronic health record (EHR) integration and geocoded social

determinants. While robust standard errors partially address clustering, our inability to definitively link multiple admissions per patient may underestimate variance for frequent utilizers. Cost data reflect charges rather than actual payments, potentially overstating absolute amounts though relative differences remain valid for comparison purposes. The 2016–2020 timeframe, while providing pre-pandemic baseline data, may not fully reflect post-COVID healthcare patterns; however, sensitivity analyses excluding 2020 data showed minimal impact on primary findings. Finally, findings from Southeastern Virginia may have limited generalizability to other regions, though this geographic specificity strengthens our ability to develop targeted local interventions through the HRBRC initiative.

## Conclusion

This comprehensive analysis of diabetes hospitalization costs among Southeastern Virginia's African American and Hispanic populations fundamentally challenges how we understand and manage diabetes in minority communities. Our most striking finding, that urgent admissions for patients with diabetes without major complications or comorbidities trigger a 239.5% cost increase, represents not merely a statistical anomaly but a critical healthcare system failure. These young adults, averaging just 38.7 years, experience metabolic crises requiring intensive intervention despite lacking the traditional markers of complexity that would flag them as high-risk. This paradox reveals that our current classification systems and care delivery models are fundamentally inadequate for minority populations where social determinants of health may be more predictive of crisis than clinical markers.

The readmission paradox identified in our analysis further underscores the need to reconceptualize quality metrics for complex diabetes care. The 5.8% cost reduction associated with readmission for patients with major complications suggests that staged care strategies may represent optimal management rather than system failure. Combined with our finding that minority patients develop major complications 7–10 years earlier than national averages, these patterns demand a complete reimagining of diabetes care delivery that acknowledges the unique trajectory of disease progression in underserved populations.

Our robust predictive models, achieving $R^2$ values of 0.779 to 0.789 for complicated diabetes categories, establish the strong foundation needed for the HR Biomedical Research Consortium funded ML platform. The identification of urgent admission status as the most powerful cost predictor, exceeding even complex comorbidities and major procedures, provides critical insight for algorithm development. By incorporating these findings into predictive models, we can identify patients at risk for metabolic crisis before they present urgently, potentially preventing both human suffering and the extreme financial burden these events create.

The implications of this research extend far beyond Southeastern Virginia. Healthcare systems nationwide serving minority populations must recognize that traditional approaches to diabetes management fail to capture the true drivers of cost and crisis. The youngest patients may incur the highest costs, seemingly uncomplicated cases may require the most intensive resources, and planned readmissions may represent sophisticated care rather than failure. These counterintuitive patterns reflect the complex interplay of clinical disease, social determinants, and systemic barriers that characterize diabetes in minority populations.

Moving forward, several urgent actions are required. First, healthcare systems must develop rapid-access diabetes stabilization programs specifically targeting young minority adults, recognizing that urgent presentation often represents the culmination of multiple access barriers rather than sudden disease progression. Second, screening and prevention programs must begin in the third decade of life for minority populations, a full decade earlier than current guidelines suggest. Third, reimbursement models must evolve to recognize that diabetes complexity cannot be captured by presence or absence of documented complications alone; social determinants and access barriers may be equally important in predicting resource needs.

The extreme cost disparities identified in this study are not inevitable. They represent opportunities for targeted intervention that could dramatically improve both health outcomes and healthcare sustainability. The HRBRC-funded

prediction platform under development will enable proactive identification of high-risk patients, real-time monitoring of cost drivers, and targeted resource allocation to prevent crises before they occur. By transforming our approach from reactive crisis management to predictive prevention, we can begin to address the profound disparities that have made diabetes a disease of inequity.

This study provides compelling evidence that diabetes in minority populations defies conventional categorization and requires innovative approaches to prediction, prevention, and management. The 239.5% cost increase for urgent admissions in patients without major complications should serve as a clarion call for immediate action. Only through sophisticated predictive analytics, targeted intervention strategies, and fundamental restructuring of care delivery can we hope to address the diabetes crisis devastating minority communities. At the same time, paradoxical findings such as the cost reduction with readmission in complex cases underscore that future research is urgently needed to disentangle these dynamics, clarify underlying mechanisms, and refine strategies for equitable care.The time for incremental change has passed; the magnitude of disparities revealed here demands transformation.

### Implications for policy and practice

The findings from this study have immediate implications for healthcare delivery, policy development, and research priorities in minority-serving health systems.

The 239.5% cost increase associated with urgent admissions in patients without major complications mandates the establishment of diabetes rapid-response programs in all emergency departments, with particular priority in systems serving high proportions of minority patients. Healthcare systems should implement risk stratification protocols that incorporate age under 40 and urgent presentation as high-risk indicators regardless of documented complications. The paradoxical cost reduction with readmission for DMCC patients (5.8%) suggests that planned readmission protocols for complex diabetes patients could improve both outcomes and costs.

For Policy Makers: Current MS-DRG-based reimbursement systems fail to capture the true cost variability within diabetes categories, particularly for DWO patients where costs can vary from minimal to extreme. Medicare and Medicaid programs should adjust risk stratification to account for the interaction between age, admission type, and social determinants that drive extreme costs in minority populations. Quality metrics penalizing all readmissions should be reconsidered for patients with multiple major complications where staged care may represent optimal management.

For Research Priorities: This study establishes the foundation for the HRBRC-funded ML prediction platform, with baseline $R^2$ values of 0.779–0.789 providing strong evidence for feasibility of enhanced prediction. Future research should focus on integrating social determinants data, developing real-time risk prediction algorithms, and testing intervention strategies targeting young adults at risk for metabolic crises. The extreme cost drivers identified here should guide feature selection in predictive models and inform development of early warning systems for diabetes decompensation.

For Community Health Programs: The finding that patients with diabetes develop major complications by age 54.9 years in our minority population, compared to national averages in the seventh decade, necessitates community screening programs beginning at age 30 rather than 45. Education programs must address the specific risk of urgent metabolic crises in young adults with diabetes, emphasizing that absence of documented complications does not equate to low risk.

### Supporting information

**S1 Table. Continuous Predictors of Healthcare Costs: Effect Size per Unit Increase.** ***p < 0.001, **p < 0.01, *p < 0.05
Note: Coefficients represent percentage change in charges per unit increase.
(DOCX)

## Acknowledgments

We extend our heartfelt gratitude to the Hampton Roads Biomedical Research Consortium (HRBRC) for their generous financial support of this research project. Their commitment to advancing medical research and addressing health disparities in the Hampton Roads region has made this study possible. We acknowledge their ongoing support and dedication to fostering scientific discovery and improving community health outcomes. VHI has provided non-confidential patient level information used in this study which it has compiled in accordance with Virginia Law but which it has no authority to independently verify. By using this data, the authors agree to assume all risks that may be associated with or arise from the use of inaccurate data. VHI cannot and does not represent that the use of VHI's data was appropriate for this study or endorse or support any conclusions or inferences that may be drawn from the use of VHI's data. This work was conducted in conjunction with M. Foscue Brock Institute for Community and Global Health at Macon & Joan Brock Virginia Health Sciences at Old Dominion University.

## Author contributions

**Conceptualization:** Ismail EL Moudden, Qi Zhang.

**Data curation:** Ismail EL Moudden, Michael C. Bittner.

**Formal analysis:** Ismail EL Moudden, Michael C. Bittner.

**Funding acquisition:** Ismail EL Moudden, Reem Sharaf-Alddin, Michael C. Bittner, Qi Zhang.

**Investigation:** Ismail EL Moudden, Reem Sharaf-Alddin, Qi Zhang.

**Methodology:** Ismail EL Moudden, Reem Sharaf-Alddin, Qi Zhang.

**Project administration:** Ismail EL Moudden, Asra Amidi, Michael C. Bittner.

**Resources:** Ismail EL Moudden, Asra Amidi, Qi Zhang.

**Software:** Ismail EL Moudden, Michael C. Bittner.

**Supervision:** Ismail EL Moudden, Qi Zhang.

**Validation:** Ismail EL Moudden, Qi Zhang.

**Visualization:** Ismail EL Moudden.

**Writing – original draft:** Ismail EL Moudden.

**Writing – review & editing:** Ismail EL Moudden, Reem Sharaf-Alddin, Asra Amidi, Michael C. Bittner, Qi Zhang.

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
