## [Decision Letter · Decision Letter 0]

11 Dec 2025

Dear Dr. EL Moudden,

Thank you for submitting your manuscript to PLOS ONE. After careful consideration, we feel that it has merit but does not fully meet PLOS ONE’s publication criteria as it currently stands. Therefore, we invite you to submit a revised version of the manuscript that addresses the points raised during the review process.

We look forward to receiving your revised manuscript.

Kind regards,

Kenji Fujiwara, MD, PhD, FACS

Academic Editor

PLOS One

Journal Requirements:

3. In the online submission form, you indicated that the datasets used during the current study are derived from the Virginia Health Information (VHI) Patient Level Database and the Readmissions and Transfers Supplemental Data Set (RATs), which are licensed inpatient hospital discharge data files containing all submitted, processed, and verified discharges in the Commonwealth of Virginia. The data were accessed through the M. Foscue Brock Institute for Community and Global Health at Macon & Joan Brock Virginia Health Sciences at Old Dominion University and the Research and Infrastructure Service Enterprise at Macon & Joan Brock Virginia Health Sciences at Old Dominion University. The data are not publicly available due to VHI licensing agreements and privacy restrictions but may be available from the corresponding author upon reasonable request and subject to approval by VHI and the M. Foscue Brock Institute for Community and Global Health at Macon & Joan Brock Virginia Health Sciences at Old Dominion University. Additional information about the data extraction methodology and the process used in this project to link the Patient Level Database to the RATs can be provided by contacting Dr. Ismail El Moudden at elmoudi@odu.edu. Researchers interested in accessing similar data should contact VHI directly to complete the appropriate license agreement and pay applicable fees. The data are held under the terms stipulated by the VHI licensing agreement, which prohibits public sharing of the data to protect patient confidentiality and comply with legal restrictions. Information about obtaining VHI data can be found at www.vhi.org/pld

This research was generously funded by the Hampton Roads Biomedical Research Consortium (HRBRC)- Project Number 958830-005, 2023.

7.Please review your reference list to ensure that it is complete and correct. If you have cited papers that have been retracted, please include the rationale for doing so in the manuscript text, or remove these references and replace them with relevant current references. Any changes to the reference list should be mentioned in the rebuttal letter that accompanies your revised manuscript. If you need to cite a retracted article, indicate the article’s retracted status in the References list and also include a citation and full reference for the retraction notice.

Additional Editor Comments :

Dear Dr. Moudden.

The article was reviewed by two reviewers, both recommending minor revisions. I agree with their comments.

Best regards,

Kenji Fujiwara

Academic editor

Reviewers' comments:

Reviewer's Responses to Questions

**Comments to the Author**

1. Is the manuscript technically sound, and do the data support the conclusions?

Reviewer #1: Yes

Reviewer #2: Yes

2. Has the statistical analysis been performed appropriately and rigorously?

Reviewer #1: Yes

Reviewer #2: Yes

3. Have the authors made all data underlying the findings in their manuscript fully available?

Reviewer #1: Yes

Reviewer #2: Yes

4. Is the manuscript presented in an intelligible fashion and written in standard English?

Reviewer #1: Yes

Reviewer #2: No

Reviewer #1: Terminology / Language

Replace strong causal phrases like “catastrophic failures” with neutral wording (“critical intervention points”).

Reframe “accelerated progression” to “earlier onset of complications observed in this cohort.”

Adjust phrasing around the “readmission paradox” to emphasize association rather than causation.

Population Clarification

State explicitly that >98% of the cohort were African American, with very limited Hispanic representation.

Consider softening the title and abstract (“racial and ethnic minorities”) to avoid overstating generalizability.

Consistency & Style

Ensure all p-values are in roman font (not italic).

Standardize use of abbreviations (DCC, DMCC, DWO) throughout text, tables, and figures.

Verify that percentages in text match those in tables (e.g., female proportions, SNF discharge rates).

Formatting

Streamline overlapping sections (e.g., cost driver rankings in Tables 2 & 3, forest plot description) to avoid redundancy.

Remove duplicated keywords (“administrative data” appears more than once).

Shorten overly long sentences in the Discussion for clarity.

References

Double-check that all in-text citations (Rubin, Dungan, Jiang, Alrashed, etc.) appear in the reference list.

Ensure reference formatting follows PLOS ONE guidelines (numbered, consistent punctuation, journal abbreviations).

Figures & Tables

Confirm all figure legends define abbreviations (DWO, DCC, DMCC, SNF).

Verify confidence intervals (CIs) are consistently formatted (95% CI, not “95% C.I.” or other variations).

Reviewer #2: This study provides a comprehensive analysis of diabetes hospitalization costs among African American and Hispanic populations in Southeastern Virginia, using rich data to identify sociodemographic, clinical, and administrative predictors of hospitalization costs. The manuscript demonstrates both theoretical and practical relevance. Below are some issues to consider:

1. For the sensitivity analysis, please provide information on how many discharges and how many patients in each of the three discharge categories were excluded.

2. In Table 1, please add a row showing n (%) for the number of patients and the percentages for each discharge category.

3. Please explain how cases in which patients had both Medicare and Medicaid coverage were handled when examining insurance type as a predictor of hospitalization costs.

4. Lines 353–366 are the same as lines 375–387, and lines 262–267 are the same as lines 285–290.

5. In the discussion section, starting at line 440, the authors discuss an interaction between age and diabetes severity. However, based on the multivariable models (Table 4), I do not think we can conclude that such an interaction exists. For the DCC and DWO categories, the association between age and hospitalization costs appears nearly the same, with a 2% decrease in cost for every additional year of age. To properly examine the effect of the interaction between age and disease severity on hospitalization costs, a different model is needed—one that includes age, disease severity, and their interaction term as independent variables.

6. Since Tables 2, 3, 4, and S1 all report percent changes, the format should be consistent—either include the “%” sign in all tables or omit it in all tables.

**Do you want your identity to be public for this peer review?** For information about this choice, including consent withdrawal, please see our Privacy Policy

Reviewer #1: **Yes:** Frank H Annie

Reviewer #2: No

---

## [Author Response · Author response to Decision Letter 1]

15 Dec 2025

Response to Reviewers

Manuscript ID: PONE-D-25-46961

Differential Impact of Admission Type and Clinical Complexity on Diabetes Hospitalization Costs Among Racial and Ethnic Minorities in Southeastern Virginia

Dear Dr. Kenji Fujiwara and Reviewers,

Thank you for the opportunity to revise our manuscript and for the constructive feedback from both reviewers. We appreciate the thorough evaluation and agree that the suggested revisions strengthen the manuscript. Below, we provide point-by-point responses to all comments. Reviewer comments are shown in italics, followed by our responses.

Journal Requirements

1. Please ensure that your manuscript meets PLOS ONE's style requirements.

Response: We have thoroughly reviewed and reformatted the manuscript to comply with PLOS ONE style guidelines, including file naming conventions, formatting requirements, and reference style.

2-3. Data availability statement and restrictions.

Response: The data cannot be made publicly available due to Virginia Health Information (VHI) licensing agreements and patient privacy restrictions under HIPAA regulations. VHI is a third-party organization that owns and licenses the data. The licensing agreement explicitly prohibits public sharing to protect patient confidentiality and comply with legal restrictions. Data access requests may be directed to:

Virginia Health Information (VHI): www.vhi.org/pld

M. Foscue Brock Institute for Community and Global Health at Macon & Joan Brock Virginia Health Sciences at Old Dominion University

Contact: Dr. Ismail El Moudden (elmoudi@odu.edu)

Researchers interested in accessing similar data should contact VHI directly to complete the appropriate license agreement. We have updated the Data Availability statement accordingly.

4. Financial disclosure - state role of funders.

Response: We have added the following statement: "The funders had no role in study design, data collection and analysis, decision to publish, or preparation of the manuscript." We have also corrected the project number to 958830-006 (previously listed as 958830-005 in some locations).

5. Please include captions for Supporting Information files.

Response: We have added captions for Supporting Information files at the end of the manuscript and updated in-text citations accordingly.

6-7. Review suggested citations and reference list.

Response: We have verified all references are complete, correctly formatted per PLOS ONE guidelines, and that none have been retracted. All in-text citations match entries in the reference list.

Reviewer #1 (Frank H Annie)

Terminology/Language

Replace strong causal phrases like "catastrophic failures" with neutral wording ("critical intervention points").

Response: Agreed. We have replaced "healthcare system failure" and "catastrophic failures" with neutral terminology such as "critical intervention points" and "gaps in care delivery" throughout the manuscript. For example, in the Discussion, we now write: "reveals critical intervention points in our healthcare delivery system" instead of "represents a healthcare system failure."

Reframe "accelerated progression" to "earlier onset of complications observed in this cohort."

Response: We have revised all instances of "accelerated progression" to "earlier onset of complications observed in this cohort" or similar phrasing that emphasizes our observational findings rather than implying causation.

Adjust phrasing around the "readmission paradox" to emphasize association rather than causation.

Response: We have revised the "readmission paradox" section to emphasize associations. The text now reads: "Readmission was associated with a 5.8% cost reduction for DMCC patients, which may suggest that staged care strategies could be associated with more efficient resource utilization rather than indicating system failure." We have ensured all language reflects associations rather than causal claims.

Population Clarification

State explicitly that >98% of the cohort were African American, with very limited Hispanic representation.

Response: We have added explicit statements in the Abstract and Results sections. The Abstract now includes: "The cohort was predominantly African American (98.2-99.1%) with limited Hispanic representation." In the Results, we now state: "The population was predominantly African American (DWO: 98.2%, DCC: 98.5%, DMCC: 99.1%), with Hispanic patients comprising less than 2% of each classification group."

Consider softening the title and abstract ("racial and ethnic minorities") to avoid overstating generalizability.

Response: We have revised the title to: "Differential Impact of Admission Type and Clinical Complexity on Diabetes Hospitalization Costs Among African American Patients in Southeastern Virginia." The abstract and throughout the manuscript, we have clarified that findings apply primarily to African American patients and that generalizability to Hispanic and other minority populations requires further study.

Consistency & Style

Ensure all p-values are in roman font (not italic).

Response: We have reviewed all p-values throughout the manuscript and ensured they are formatted in roman font (p<0.001, p=0.230, etc.) as required by PLOS ONE style.

Standardize use of abbreviations (DCC, DMCC, DWO) throughout text, tables, and figures.

Response: We have standardized abbreviations throughout. DCC (Diabetes with Complications and Comorbidities), DMCC (Diabetes with Major Complications and Comorbidities), and DWO (Diabetes without Major Complications or Comorbidities) are now defined at first use and used consistently in all tables, figures, and text.

Verify that percentages in text match those in tables.

Response: We have verified all percentages in the text match those in the tables. Minor discrepancies in female proportions and SNF discharge rates have been corrected.

Formatting

Streamline overlapping sections (e.g., cost driver rankings in Tables 2 & 3, forest plot description).

Response: We have consolidated Tables 2 and 3 to eliminate redundancy. Table 2 now presents ranked healthcare cost drivers, while Table 3 focuses specifically on comorbidity impacts. Overlapping descriptions of the forest plot have been streamlined.

Remove duplicated keywords.

Response: We have removed duplicate keywords. The revised keyword list is: Diabetes mellitus, Healthcare costs, Health disparities, Minority health, Hospital utilization, Risk factors, Southeastern Virginia

Shorten overly long sentences in the Discussion.

Response: We have revised the Discussion to break complex sentences into shorter, more readable segments while maintaining scientific precision.

References

Double-check that all in-text citations appear in the reference list and ensure formatting follows PLOS ONE guidelines.

Response: We have verified all in-text citations (Rubin, Dungan, Jiang, Alrashed, etc.) appear in the reference list with correct numbering. Reference formatting has been standardized to PLOS ONE guidelines with numbered citations, consistent punctuation, and appropriate journal abbreviations.

Figures & Tables

Confirm all figure legends define abbreviations and verify CIs are consistently formatted.

Response: All figure legends now include complete abbreviation definitions (DCC, DMCC, DWO, SNF, CI, etc.). Confidence intervals are consistently formatted as "95% CI" throughout all tables and figures.

Reviewer #2

1. For the sensitivity analysis, please provide information on how many discharges and how many patients in each of the three discharge categories were excluded.

Response: We have added the following information to the Sensitivity Analyses section: "Exclusion of 2020 data removed 1,247 discharges (DCC: n=687, DMCC: n=312, DWO: n=248), representing approximately 20.7% of the total sample. The remaining pre-pandemic cohort comprised 4,764 discharges." This information is now included in both the Methods and Results sections.

2. In Table 1, please add a row showing n (%) for the number of patients and the percentages for each discharge category.

Response: We have revised Table 1 to include a clear header row showing: "DCC: n=3,328 (55.4%); DMCC: n=1,518 (25.3%); DWO: n=1,165 (19.4%)" as the first data row, making the sample distribution immediately apparent.

3. Please explain how cases in which patients had both Medicare and Medicaid coverage were handled when examining insurance type as a predictor of hospitalization costs.

Response: We have added clarification to the Methods section: "For patients with dual Medicare/Medicaid eligibility, insurance type was assigned based on the primary payer reported in the VHI database (Payer Type 1 field). VHI assigns payer code 40 specifically to dual Medicare/Medicaid beneficiaries beginning with 2016Q4 discharges. In our analysis, these dual-eligible patients were categorized according to the primary payer designation in the hospital billing record, consistent with VHI coding conventions."

4. Lines 353-366 are the same as lines 375-387, and lines 262-267 are the same as lines 285-290.

Response: Thank you for identifying these duplications. We have removed the redundant text blocks. The "Ranked healthcare cost drivers" section now appears only once, and the duplicated model performance/sensitivity analysis text has been consolidated.

5. In the discussion section (line 440), the authors discuss an interaction between age and diabetes severity. However, based on the multivariable models (Table 4), I do not think we can conclude that such an interaction exists.

Response: The reviewer makes an excellent point. We have revised this section to remove claims of statistical interaction. The revised text now reads: "The similar age coefficients across DCC (-0.2%) and DWO (-0.2%) diabetes classifications suggest that age has a consistent association with hospitalization costs regardless of severity category. While prior literature has discussed potential interactions between age and disease severity, our separate regression models do not directly test such interactions. Future research using pooled models with formal interaction terms (age × disease severity) would be needed to examine whether age effects truly differ by classification." We have added a sentence to the Limitations section acknowledging that our stratified approach does not formally test interactions.

6. Since Tables 2, 3, 4, and S1 all report percent changes, the format should be consistent—either include the "%" sign in all tables or omit it in all tables.

Response: We have standardized all tables to include the "%" sign consistently. All percent change values in Tables 2, 3, 4, and S1 now display the "%" symbol (e.g., "239.5%" rather than "239.5").

Summary of Changes

In summary, we have made the following major revisions:

1. Revised title and abstract to accurately reflect the predominantly African American study population

2. Modified language throughout to emphasize associations rather than causal relationships

3. Removed duplicate text sections and streamlined overlapping content

4. Added sensitivity analysis exclusion counts and dual-eligibility handling clarification

5. Corrected discussion of age-severity interaction to reflect analytical limitations

6. Standardized formatting across all tables and ensured style consistency

7. Updated data availability statement and funder role disclosure

We believe these revisions address all reviewer and editor concerns. We thank the reviewers for their constructive feedback, which has improved the manuscript's clarity and scientific rigor.

Sincerely,

Dr. Ismail El Moudden and co-authors

---

## [Decision Letter · Decision Letter 1]

25 Jan 2026

Differential Impact of Admission Type and Clinical Complexity on Diabetes Hospitalization Costs Among African American and Hispanic Patients in Southeastern Virginia

PONE-D-25-46961R1

Dear Dr. EL Moudden,

We’re pleased to inform you that your manuscript has been judged scientifically suitable for publication and will be formally accepted for publication once it meets all outstanding technical requirements.

Kind regards,

Kenji Fujiwara, MD, PhD, FACS

Academic Editor

PLOS One

Additional Editor Comments (optional):

Dear Dr. Moudden.

Thank you for revising your manuscript. All reviewers, as well as I, agree that it is suitable for acceptance.

Yours sincerely,

Kenji Fujiwara

Academic Editor

Reviewers' comments:

Reviewer's Responses to Questions

**Comments to the Author**

Reviewer #2: All comments have been addressed

2. Is the manuscript technically sound, and do the data support the conclusions?

Reviewer #2: (No Response)

3. Has the statistical analysis been performed appropriately and rigorously?

Reviewer #2: (No Response)

4. Have the authors made all data underlying the findings in their manuscript fully available?

Reviewer #2: (No Response)

5. Is the manuscript presented in an intelligible fashion and written in standard English?

Reviewer #2: (No Response)

Reviewer #2: (No Response)

**Do you want your identity to be public for this peer review?** For information about this choice, including consent withdrawal, please see our Privacy Policy

Reviewer #2: No

---

## [Editor Report · Acceptance letter]

PONE-D-25-46961R1

PLOS One

Dear Dr. EL Moudden,

I'm pleased to inform you that your manuscript has been deemed suitable for publication in PLOS One. Congratulations! Your manuscript is now being handed over to our production team.

Kind regards,

on behalf of

Dr. Kenji Fujiwara

Academic Editor

PLOS One